# An Online-Based Survey to Assess Knowledge, Attitudes, and Barriers to Perform First Aid after Road Accidents Conducted among Adult Jordanians

**DOI:** 10.3390/healthcare12090947

**Published:** 2024-05-05

**Authors:** Walid Al-Qerem, Anan Jarab, Abdel Qader Al Bawab, Alaa Hammad, Judith Eberhardt, Fawaz Alasmari, Haneen Kalloush, Lujain Al-Sa’di, Raghd Obidat

**Affiliations:** 1Department of Pharmacy, Faculty of Pharmacy, Al-Zaytoonah University of Jordan, Amman 11733, Jordan; abdelqader.albawab@zuj.edu.jo (A.Q.A.B.); alaa.hammad@zuj.edu.jo (A.H.); haneen.kalloush@zuj.edu.jo (H.K.); phres01@zuj.edu.jo (L.A.-S.); phres02@zuj.edu.jo (R.O.); 2College of Pharmacy, Al Ain University, Abu Dhabi 112612, United Arab Emirates; asjarab@just.edu.jo; 3AAU Health and Biomedical Research Center, Al Ain University, Abu Dhabi 112612, United Arab Emirates; 4Department of Clinical Pharmacy, Faculty of Pharmacy, Jordan University of Science and Technology, Irbid 22110, Jordan; 5Department of Psychology, School of Social Sciences, Humanities and Law, Teesside University, Borough Road, Middlesbrough TS1 3BX, UK; j.eberhardt@tees.ac.uk; 6Department of Pharmacology and Toxicology, College of Pharmacy, King Saud University, Riyadh 12372, Saudi Arabia; ffalasmari@ksu.edu.sa

**Keywords:** first aid, Jordan, knowledge, attitude, barriers

## Abstract

(1) Background: First aid administered during road accidents can save millions of lives. However, the knowledge and attitudes of the Jordanian population towards first aid are lacking. This study aimed to examine the knowledge, attitudes, and barriers to performing first aid among the Jordanian population during road accidents. (2) Methods: An online questionnaire was developed and distributed using various Jordanian social media platforms. The questionnaire collected the participants’ sociodemographic details and assessed their first aid knowledge, attitudes toward first aid, and barriers preventing the participants from performing first aid in emergencies. (3) Results: 732 participants participated in this study. The median knowledge score regarding first aid items was 9 (7–10) out of the maximum possible score of 15. The median first aid attitude score was 24 (22–27) out of a maximum possible score of 30. The most commonly reported barrier to performing first aid among the participants was “lack of first aid training” (76.78%), followed by “lack of knowledge about first aid” (75.81%) and “fear of performing first aid” (57.51%). The participants with lower income levels exhibited more negative attitudes towards first aid (4). Conclusions: This study underscores the urgent need for enhanced first aid training and awareness in Jordan. The participants’ first-aid knowledge overall was limited, although positive attitudes toward first-aid delivery were observed. The findings emphasize the need for regular and structured first-aid training courses, addressing barriers such as fear and misinformation and ensuring accessibility across all socioeconomic levels to improve preparedness for road traffic accidents and other emergencies. This comprehensive approach can better equip the Jordanian population to effectively manage emergencies and improve public health outcomes.

## 1. Introduction

First aid is identified as the interventions and assessments that can be carried out by a bystander or the victim themselves using minimal or no first aid supplies [1,2,3]. First aid is administered to individuals who are sick or injured in critical and life-threatening situations to prevent further deterioration, help in the treatment process, or sustain victims until professional medical assistance can be obtained [4]. A first-aid provider is an individual trained in first aid, medicine, or emergency care [1]. According to European Resuscitation Council Guidelines, first aid is defined as “the initial care provided for an acute illness or injury to preserve life, alleviate suffering, prevent further illness or injury and promote recovery” [5], while basic life support (BLS) is more concerned with “initiating CPR in any unresponsive person with absent or abnormal breathing”. This study mainly focused on investigating first aid components [6].

According to the World Health Organization (WHO), road traffic crashes contribute globally to about 1.19 million deaths annually. More than 3500 individuals worldwide lose their lives in road accidents daily. Furthermore, 20 to 50 million individuals sustain non-fatal injuries, with many experiencing disabilities as a result of traffic accidents. Moreover, road traffic injuries are considered to be a leading cause of death for individuals aged 15–29 [7]. Traffic accidents are a significant health concern in Jordan and rank as the second-leading cause of death in the country. Data collected between 1998 and 2007 indicate significant increases in road traffic accidents [8]. Moreover, approximately 562 deaths and 17,096 injuries were caused by traffic accidents in Jordan in 2022 alone [9].

A meta-analysis review conducted in 2018 reported that bystander cardiopulmonary resuscitation enhances out-of-hospital cardiac arrest survival [10]. Based on a study conducted in the United States, bystander CPR was found to be associated with higher in-hospital survival rates among older patients with out-of-hospital cardiac arrest [11]. These findings highlight the life-saving potential of widespread BLS first-aid education [12]. However, the levels of this knowledge are limited. For example, a cross-sectional study conducted in South India aimed to assess first-aid knowledge among medical students using a self-administered questionnaire. The findings indicated that the majority of the participants lacked substantial first-aid knowledge, highlighting the need for formal first-aid training [13]. Similarly, a cross-sectional study assessing first-aid knowledge among 1500 university students in Jordan showed that their levels of first-aid knowledge were inadequate, and a systematic practical first-aid course was recommended at the secondary school level [14]. Few studies have been published on first-aid knowledge in the Middle East [4,15,16,17].

To date, no published studies have evaluated knowledge, attitudes, and barriers to performing first aid in Jordan among the general population. Therefore, the present study aimed to investigate knowledge, attitudes, and barriers to performing first aid during road accidents within the Jordanian population. This study will also develop an Arabic tool to evaluate knowledge, attitudes, and barriers to performing first aid, which could be used in different Arabic-speaking countries.

## 2. Materials and Methods

An online questionnaire was developed using Google Forms based on a thorough literature review [18,19,20]. The questionnaire link and summary of the study objectives were distributed using various Jordanian social media platforms. Responses were collected between February and March 2024. The inclusion criteria consisted of being a Jordanian resident aged 18 or older. To confirm that the responds met the inclusion criteria, questions about place of residency, nationality, and age were included in the questionnaire. The authors obtained ethical approval from the Institutional Review Board and the Deanship of Research at Al-Zaytoonah University of Jordan. This study followed the Declaration of Helsinki ethical guidelines. Ethical approval was secured from Al-Zaytoonah University of Jordan on 10 September 2022 (Ref#19/09/2022–2023).

### 2.1. Data Collection and Study Instruments

The questionnaire contained six sections. The first section inquired about participants’ possession of a driving license. If a participant answered “no”, they would be automatically redirected to the third section, skipping the second section. The second section focused on the participants’ driving history, encompassing four items: participants’ occupation, the number of years they had been driving, their daily driving hours, and whether they owned a first aid kit. The third section covered the participants’ sociodemographic profile, including their gender, age, educational level, monthly income level, marital status, whether the participant had any children, sleeping hours, estimated number of accidents witnessed in the past year, prior first-aid training, and awareness of the Jordanian emergency response number. The fourth section evaluated the participants’ first-aid knowledge. The knowledge score was calculated by awarding a point for every correct response, with a maximum score of 15 points. The knowledge section was divided into two parts: the first comprised 7 items focusing on general first aid knowledge, while the second consisted of 8 items representing the perceived components of first aid. The fifth section contained six items assessing the participants’ attitudes towards providing first aid, and the final section was dedicated to identifying the barriers that the participants faced to performing first aid measures.

### 2.2. Tool Validation

The content validity of the questionnaire was evaluated by a group of experts, including three first-aid experts. The questionnaire was initially developed in English, in line with the literature review, which had also been conducted in English. The questionnaire underwent a forward–backward translation process into Arabic by different translators, given that Arabic was the native language of the study sample. A pilot study involving 30 Jordanian individuals was conducted to ensure the questionnaire’s suitability and clarity for Jordanian participants. The pilot study data were not included in the statistical analysis. In addition, the internal consistency and reliability of the two generated scales assessing knowledge and attitudes were evaluated using Cronbach’s alpha. For the attitude scale, an acceptable Cronbach’s alpha value was set at >0.7, whereas for the knowledge scale, it was deemed acceptable if it was greater than 0.5 [21], given that lower Cronbach’s alpha values are anticipated with binary data [22]. The Cronbach’s alpha values obtained in the present study were acceptable (0.52 and 0.92).

### 2.3. Sample Size Calculations

Convenience and snowball sampling methods were adopted [23]. The Krejcie and Morgan formula was used to calculate the minimum required sample size:S = X2NP(1 − P) + a2(N − 1) + X2P(1 − P)
where S = required sample size, X2 = the table value of chi-square for 1 degree of freedom at the desired confidence level (3.841), N = the population size, P = the population proportion (assumed to be 50 to provide the maximum sample size), and d = the degree of accuracy expressed as a proportion (0.5). Krejcie and Morgan formulated predefined sample size tables based on the above formula. In the current study, an indefinite population was considered to determine the required sample size. Consequently, the minimum sample size needed was 384 for a 95% confidence level (CI) and a 5% margin of error.

### 2.4. Statistical Analysis

The Statistical Package for the Social Sciences (SPSS) version 26.0 was used for data analysis. Categorical variables were presented using percentages and frequencies, while continuous variables were described using the median and the 25th to 75th percentiles. As the data were not normally distributed, nonparametric tests were conducted. Two quantile regression models were applied to assess the predictors’ association with knowledge and attitude scores. The predictors included age, sex, income, education levels, marital status, parental status, possession of a driving license, the number of accidents witnessed in the past year, and prior first aid training. The knowledge score was included as an additional predictor in the attitude regression model. The significance level was established at a threshold of *p* < 0.05.

## 3. Results

Table 1 displays the participants’ sociodemographic profile. The present study included 732 participants (66.1%female). The median age was 24 years (ranging from 22 to 30), and most of the participants were single (75.3%). Most earned 500 to 1000 Jordanian Dinars [JOD] (44.4%), and more than half had no children (78.3%). The majority reported getting 6 to 8 h of sleep daily (70.9%). More than half of the participants had a driving license (59.3%). The most frequently reported number of road accidents witnessed in the past year was one to five accidents (85.2%). The vast majority of the participants reported knowing the emergency response number (91.8%). Finally, more than half of the participants had prior first-aid training (54.2%).

Table 2 shows the demographic characteristics of the participants with a driving license. Most were regular drivers (95.7%), and many had been driving for less than 5 years (33.9%). Most spent 1 to 3 h driving daily (50.1%), and most did not have a first aid kit in their cars (66.8%).

The participants’ responses to the general knowledge items regarding performing first aid are presented in Table 3. The most correctly answered item was “The place where first aid should be initiated” (95.3%), followed by the item “First aid for breaking a limb” (53.4%), while the least-correctly answered item was “Which of the following are airway maneuvers” (3.0%).

The participants’ responses to the first aid items are displayed in Table 4. The most frequently correctly selected response was “Making sure that the patient is breathing properly” (96.8%), followed by “Stopping bleeding” (91.7%), while the least-frequently correctly selected response was “Giving fluid to drink” (23.9%).

The participants’ median knowledge score in relation to first aid was 9 (7–10) out of the maximum possible score of 15.

Table 5 displays the participants’ attitudes toward first aid. Most of the participants agreed/strongly agreed with the item “First aid can help save lives” (86.8%), followed by the item “First aid is important” (86.4%), but most disagreed/strongly disagreed with the item “I am willing to provide first aid” (14.9%). The participants’ median first-aid attitude score was 24 (22–27) out of a maximum possible score of 30. The scale showed high internal consistency, with a Cronbach’s alpha of 0.92.

The participants’ reported barriers to performing first aid are presented in Figure 1. The most-reported barrier was “lack of first aid training” (76.78%), followed by “lack of knowledge about first aid “(75.81%) and “fear of performing first aid” (57.51%), while the least-reported barrier was “for legal reasons” (31.28%).

Two quantile regression models were applied to identify the variables that were significantly associated with the participants’ knowledge and attitudes toward performing first aid (Table 6). The results showed that the participants who did not have first-aid training had significantly lower knowledge scores compared with the participants who did have first-aid training (−1.000, 95% CI (−1.428, −0.572), *p* < 0.001). Moreover, the participants who exhibited higher knowledge scores had significantly greater attitude scores (0.187, 95% CI (0.044–0.330), *p* = 0. 010). The participants who reported earning less than 500 JOD a month had significantly lower attitude scores than those who reported earning more than 1000 JOD (−1.153, 95% CI (−1.982, −0.323), *p* = 0.007).

## 4. Discussion

A significant proportion of deaths globally are caused by road traffic accidents [7]. Traffic accidents are a prominent health-related problem in Jordan, which places a significant burden on health services in addition to negative economic consequences [24,25]. First aid has been shown to reduce damage to health and deaths caused by accidents [26]. Therefore, this study aimed to determine Jordanian adults’ knowledge and attitudes toward performing first aid as well as the barriers and factors hindering first-aid delivery.

No similar recent studies have evaluated the knowledge, attitudes, and barriers to performing first aid among the Jordanian population. This study revealed that the study sample showed below-average first-aid knowledge (participants’ median score was 60%). Few of the participants were able to correctly identify the ways of opening the airway or the correct first-aid steps in the case of bleeding. These findings align with a 2014 study conducted with university students in Northern Jordan, which revealed that most were unable to perform effective first aid during emergencies [14]. Similar findings were observed among medical students in South India [13], taxi drivers in Uganda [19], drivers in the United Arab Emirates (UAE) [15], commercial inter-city drivers in Nigeria [20], the UAE general population [16], and teachers in government schools in Palestine [27]. Our findings suggest that well-designed, structured, and regular training courses are urgently required to increase the levels of first-aid knowledge among the Jordanian population and improve outcomes for victims of road traffic accidents and other emergency situations. Unfortunately, mandatory first-aid courses are not provided at schools or universities in Jordan. Moreover, although, the driving theory test content in Jordan include questions related to first aid, no practical training is offered.

Regarding the perceived components of first aid, most of the participants were unable to correctly identify that giving an accident victim food to eat is inappropriate, a finding supported by previously published literature [20].

The consistent lack of first aid knowledge across diverse populations indicates a broader issue that transcends regional and professional boundaries, highlighting a need for improved educational strategies. This study’s results suggest a significant gap in the basic life-saving skills among the Jordanian public, potentially increasing the morbidity and mortality associated with emergencies due to delayed or inappropriate first-aid responses. Furthermore, the misperception among the participants regarding inappropriate practices, such as giving food to an accident victim, highlights the critical need for comprehensive education that corrects common first-aid myths and teaches correct procedures.

Most of the participants in the present study had positive attitudes towards first aid, in line with previous findings, for example for the general public in Saudi Arabia [17], nursing and medical students in Saudi Arabia [28], and in the general population in the UAE [16].

In line with recommendations for bystanders to initiate first aid before the arrival of professional medical services [29], the findings of the current study indicate a strong consensus among individuals that assisting accident victims should not be exclusively the responsibility of healthcare workers. Additionally, a substantial majority believed in the life-saving potential of first aid. This indicates a willingness among the general population to participate more actively in emergency responses, which could be vital to improving survival rates in urgent situations. Additionally, the prevalent belief in the life-saving effectiveness of first aid emphasizes the need for enhanced training. These findings support policies that promote first-aid education and suggest integrating such training more thoroughly into community norms and practices, thereby potentially improving public health outcomes by preparing more individuals to respond effectively in emergencies.

Similar to previously published literature [30,31,32], the current study showed that the participants with first-aid training had significantly better first-aid knowledge compared to the participants without such training. Furthermore, those with relatively low monthly incomes (under 500 JOD) had more negative attitudes towards providing first aid compared to the participants with higher monthly incomes. This disparity may reflect socio-economic barriers such as access to training opportunities or a lack of awareness about the benefits of first aid, suggesting that targeted interventions are needed to address these gaps within lower-income groups.

Moreover, the participants with greater first-aid knowledge had more positive attitudes, which could be attributed to the higher confidence and skills resulting from their greater first-aid knowledge. This finding draws attention to the role of psychological readiness in emergency responses, which could be enhanced through more widespread and accessible first-aid training. Thus, improving first-aid education could not only increase knowledge but also positively influence attitudes, making individuals more likely to engage effectively in first-aid activities during emergencies.

A lack of first-aid knowledge, a lack of training, and fear of performing first aid were among the most common factors deterring the participants from performing first aid in emergencies. Similarly, a study conducted in India identified the most common obstacles to performing first aid as a lack of information on first aid, insufficient first-aid training, and fear [33]. Although fear was not the primary barrier to performing first aid in this study, worries over making mistakes and potentially harming the victim during first aid administration have been identified as the most prevalent barriers in research conducted in various countries such as Vietnam [34], Japan [35], and Hong Kong [36]. This widespread anxiety suggests a critical need for first-aid training programs not only to impart practical skills but also to focus on reducing anxiety and building confidence among potential first aiders.

To address the gaps in first-aid knowledge identified among the adult population in Jordan, the initiation of comprehensive community-based first-aid training programs is recommended. These should be a collaborative effort involving local health authorities, educational institutions, and community centers, designed to offer flexible and accessible learning opportunities for working adults. By integrating both in-person and digital platforms, these programs could cater to diverse schedules and preferences, ensuring widespread participation. Furthermore, embedding first-aid training within workplace health and safety initiatives could amplify the impact of such programs.

Incorporating targeted strategies to address the specific barriers identified, such as fear and misinformation, these programs should also include components that build confidence and dispel common first aid myths. Special attention should be paid to economically disadvantaged groups, who exhibited more negative attitudes towards first aid, by providing free or subsidized training to ensure no one is left behind. This holistic approach, which combines skill development with psychological empowerment, will not only enhance first aid knowledge and attitudes but also encourage a more responsive and resilient community that is capable of effectively managing emergencies.

### Strengths and Limitations

A strength of this study is its contribution to the scarce body of knowledge on first-aid preparedness within the Jordanian context. By shedding light on the current levels of first-aid knowledge, attitudes, and barriers among the adult population, this research provides a first step towards the development of targeted educational interventions and policy reforms aimed at enhancing first-aid competencies at a community level. This contribution is particularly significant given the lack of previous studies in this area within the region, thereby offering valuable insights for both local and regional public health strategies. Furthermore, the inclusion of a substantial sample size from diverse geographical regions in Jordan enhances the validity and generalizability of the findings. Although this study used an online questionnaire, potentially introducing selection and recall biases, previous research has demonstrated that web-based studies are an effective, efficient, and cost-effective method for generating representative samples from a population [37]. Online questionnaires offer a private and secure environment, allowing participants to respond accurately to questions. Furthermore, they facilitate access to individuals who might otherwise be difficult to reach [38]. Thus, this study not only addresses a critical gap in the existing literature but also highlights the need for initiatives to bolster first-aid readiness in Jordan.

## 5. Conclusions

The findings of the current study suggest low levels of first-aid knowledge within the adult Jordanian population, albeit with positive attitudes towards its administration. To improve outcomes in emergency situations, it is strongly advised to implement well-structured first-aid training courses and to include practical training as a requirement for obtaining a driving license and as mandatory courses for high school and university students. This recommendation is underscored by the fact that most of the participants identified a lack of first-aid training, insufficient first-aid knowledge, and fear of performing first-aid as the primary obstacles preventing them from performing first-aid when necessary.

Further, considering the significant relationship between income levels and attitudes towards first aid, it is essential to ensure that these training programs are accessible to all socio-economic segments, particularly those with lower incomes who display more negative attitudes. Additionally, the pervasive fear of making mistakes during first aid administration suggests that these training programs should not only provide practical skills but also focus on building confidence through simulation exercises and psychological support. By addressing these needs, such training initiatives can foster a more prepared and proactive community that is ready to respond effectively to emergencies and contribute to better public health outcomes. Finally, as the results indicated that the majority of the drivers did not have a first-aid kit in their motor vehicles, imposing a law that makes this compulsory will have a significant impact on saving lives during road accidents.

## Figures and Tables

**Figure 1 healthcare-12-00947-f001:**
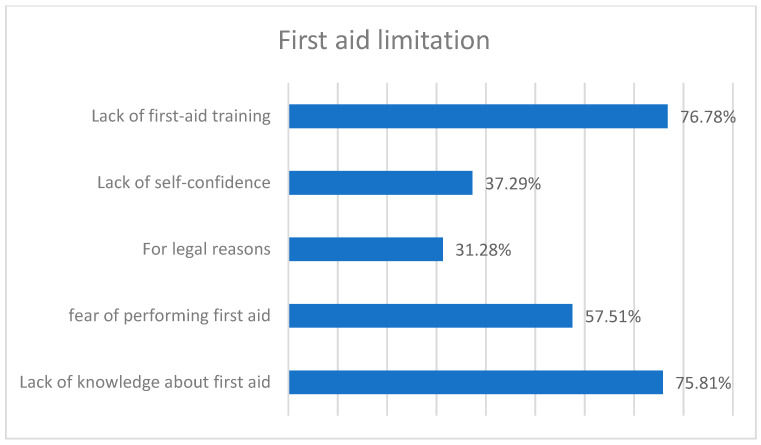
Reported barriers to performing first aid.

**Table 1 healthcare-12-00947-t001:** Sociodemographic characteristics of the participants.

	Median (Percentile 25–75)	N (%)
Age	24 (22–30)	
Sex	Female	484 (66.1)
	Male	248 (33.9)
Educational level	High school or less	267 (36.5)
	Collage/university degree	465 (63.5)
Income	Less than 500 JOD *	251 (34.3)
	500 to 1000 JOD	325 (44.4)
	More than 1000 JOD	156 (21.3)
Marital status	Single	551 (75.3)
	Married	181 (24.7)
Parental status	No	573 (78.3)
	Yes	159 (21.7)
Average sleep duration	Less than 6 h	163 (22.3)
	6 to 8 h	519 (70.9)
	More than 8 h	50 (6.8)
Possession of a driving license	No	298 (40.7)
	Yes	434 (59.3)
Estimated number of accidents witnessed in the past year	1 to 5 accidents	624 (85.2)
	More than 5 accidents	108 (14.8)
Prior first-aid training	No	335 (45.8)
	Yes	397 (54.2)
Knowledge of the emergence response number	No	60 (8.2)
	Yes	672 (91.8)

* JOD: Jordanian dinar, equivalent to 1.41 USD.

**Table 2 healthcare-12-00947-t002:** Demographic characteristics of the participants with a driving license.

		N (%)
Work as a professional driver	Professional driver (works as a driver)	19 (4.3)
	Regular driver	424 (95.7)
Driving experience	Less than a year	82 (18.5)
	Less than 5 years	150 (33.9)
	5 to 9 years	91 (20.5)
	10 years or more	120 (27.1)
Number of hours spent driving daily	Less than an hour	156 (35.2)
	1 to 3 h	222 (50.1)
	More than 3 h	65 (14.7)
Presence of a first aid kit in car	No	296 (66.8)
	I don’t know	20 (4.5)
	Yes	127 (28.7)

**Table 3 healthcare-12-00947-t003:** Frequency of responses in relation to the participants’ knowledge about first aid.

		N (%)
The place where first aid should be initiated	As soon as the victim reaches the hospital	15 (2.1)
	Immediately at the scene *	689 (95.3)
	Don’t know	19 (2.6)
Who initiates first aid?	Bystander *	264 (36.5)
	Those with knowledge of first-aid airway maneuvers only	275 (38)
	Health workers only	124 (17.2)
	Don’t know	60 (8.3)
How do you position the victim who has airway compromise?	Place the victim sideways *	350 (48.4)
	Place the victim face up	227 (31.4)
	Don’t know	146 (20.2)
First aid for broking limb	Apply a splint *	386 (53.4)
	Leave it open until the victim is in the hospital	205 (28.4)
	Don’t know	132 (18.3)
Signs of airway compromise	Fast breathing #	242 (33.06)
	Noisy breathing #	257 (35.10)
	Slow breathing #	342 (46.72)
	No breathing #	347 (47.40)
	I don’t know	100 (13.66)
Which of the following are airway maneuvers	Jaw thrust #	66 (9.01)
	Chin lift with head tilt #	459 (62.70)
	Recovery position #	149 (20.35)
	I don’t know	176 (24.04)
First aid for bleeding	Applying a tourniquet #	365 (49.86)
	Applying pressure and dressing #	401 (54.78)
	Applying alcohol	69 (9.42)
	Raising the injured body part above the body level #	265 (36.20)
	I don’t know	88 (12.02)

* correct answer, # All these options must be selected to be considered.

**Table 4 healthcare-12-00947-t004:** Participants’ knowledge about first-aid responses to road traffic accidents.

	No N (%)	Not Sure N (%)	Yes N (%)
Preventing further accidents. *	183 (25.3)	91 (12.6)	449 (62.1)
Moving patients from the accident site when required. *	101 (14)	56 (7.7)	566 (78.3)
Making sure that the patient is breathing properly. *	5 (0.7)	18 (2.5)	700 (96.8)
Stopping bleeding. *	25 (3.5)	35 (4.8)	663 (91.7)
Splinting fractures. *	301 (41.6)	92 (12.7)	330 (45.6)
Transporting patients to hospitals. *	90 (12.4)	45 (6.2)	588 (81.3)
Giving fluid to drink. **	419 (58)	131 (18.1)	173 (23.9)
Giving food to eat. **	183 (25.3)	91 (12.6)	449 (62.1)

* “Yes” is the correct answer; ** “No” is the correct answer.

**Table 5 healthcare-12-00947-t005:** Frequency of responses in relation to the participants’ attitudes towards first aid.

	Strongly Disagree N (%)	Disagree N (%)	Neutral N (%)	Agree N (%)	Strongly Agree N (%)
First aid is important	60 (8.3)	9 (1.2)	27 (3.7)	282 (39)	345 (47.7)
I am willing to provide first aid	55 (7.6)	53 (7.3)	169 (23.4)	302 (41.8)	144 (19.9)
I am willing to undergo first-aid training	50 (6.9)	24 (3.3)	54 (7.5)	338 (46.7)	257 (35.5)
First aid can help save lives	55 (7.6)	11 (1.5)	30 (4.1)	278 (38.5)	349 (48.3)
It’s easy to get first-aid training	43 (5.9)	50 (6.9)	184 (25.4)	323 (44.7)	123 (17)
I believe that helping accident victims should not be left to healthcare workers alone	57 (7.9)	37 (5.1)	111 (15.4)	324 (44.8)	194 (26.8)

**Table 6 healthcare-12-00947-t006:** Quantile regression models for demographic variables and knowledge and attitude scores towards first aid.

Parameter	Knowledge Score					Attitude Score			
	Parameter Estimates (q = 0.5)								
		Coefficient	*p*	95% Confidence Interval		Coefficient	*p*	95% Confidence Interval	
				Lower Bound	Upper Bound			Lower Bound	Upper Bound
Intercept		9.000	<0.001	7.319	10.681	22.985	<0.001	20.351	25.619
Age		−3.915 × 10^−18^	1.000	−0.036	0.036	−0.023	0.376	−0.074	0.028
Gender	Female	2.932 × 10^−15^	1.000	−0.475	0.475	0.195	0.572	−0.482	0.871
	Male	0 ^a^	.	.	.	0 ^a^	.	.	.
Education level	High school or less	3.045 × 10^−15^	1.000	−0.452	0.452	0.492	0.131	−0.147	1.132
	Collage/university degree	0 ^a^	.	.	.	0 ^a^	.	.	.
Income status	Less than 500 JOD	2.290 × 10^−15^	1.000	−0.587	0.587	−1.153	0.007	−1.982	−0.323
	500 to 1000 JOD	3.127 × 10^−15^	1.000	−0.561	0.561	−0.702	0.083	−1.496	0.092
	More than 1000 JOD	0 ^a^	.	.	.	0 ^a^	.	.	.
Marital status	Single	1.497 × 10^−15^	1.000	−0.831	0.831	0.813	0.175	−0.362	1.988
	Married	0 ^a^	.	.	.	0 ^a^	.	.	.
Estimated number of accidents witnessed last year	1 to 5 accidents	−1.213 × 10^−15^	1.000	−0.602	0.602	0.328	0.450	−0.524	1.180
	More than 5 accidents	0 ^a^	.	.	.	0 ^a^	.	.	.
Have a driving license	No	2.017 × 10^−15^	1.000	−0.496	0.496	0.164	0.646	−0.537	0.865
	Yes	0 ^a^	.	.	.	0 ^a^	.	.	.
Have kids	No	−2.981 × 10^−15^	1.000	−0.963	0.963	0.038	0.956	−1.324	1.400
	Yes	0 ^a^	.	.	.	0 ^a^	.	.	.
Have first-aid training	No	−1.000	<0.001	−1.428	−0.572	−0.573	0.067	−1.186	0.041
	Yes	0 ^a^	.	.	.	0 ^a^	.	.	.
Knowledge score						0.187	0.010	0.044	0.330

^a^: reference group.

## Data Availability

The dataset supporting the conclusions of this article is available in the Zenodo repository, https://doi.org/10.5281/zenodo.10937899.

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
