# Peer review of "An Online-Based Survey to Assess Knowledge, Attitudes, and Barriers to Perform First Aid after Road Accidents Conducted among Adult Jordanians"

_healthcare, 2024, doi:10.3390/healthcare12090947_

Round 1
Reviewer 1 Report
Comments and Suggestions for Authors
Review of the manuscript: Knowledge, attitudes, and barriers to performing first aid during road accidents among the adult Jordanian population.
The manuscript covers a very important topic for public health. Namely, the study investigated the knowledge, attitudes, and barriers to performing first aid among the Jordanian population during road accidents.
Generally, the manuscript is technically correct, and the title fully corresponds to the research objective and topic; however, the manuscript requires some corrections.
The introduction provides definitions of first aid and road traffic accidents. The authors also provide statistics on the frequency of traffic accidents in Jordan, but these are outdated. It would be good if they offered more recent data. In addition, the authors state why knowledge of first aid is important using the example of CPR, but these data date back to 1982. The authors can find confirmations for this claim in the available databases with significantly more recent references.
Given that the study was conducted in Jordan, I believe the authors should introduce potential readers to the fact that there is any if any, first aid training. For example, is it conducted at school or mandatory when taking a driving test or something similar? At the end of the introduction, the aim of the study is clearly defined. Namely, in this study, the authors wanted to investigate the Jordanian population's knowledge, attitudes, and barriers to performing first aid during road accidents.
The method does contain all the necessary elements. The authors stated study design and setting, sample and data collection, instrument, and data analysis.
The results are not clearly presented. It is necessary to correct the table for clarity. Insert a row labelled n and %, and then delete % from the cells in the table. However, only descriptive data is provided. Considering that the data analysis stated that a regression analysis was performed to determine predictors of knowledge and attitudes, it is necessary to add a table with those results.
The conclusions are based on the obtained results. The conclusion is generally drawn based on the results. The authors should be more specific in their findings and discussion when recommending first aid training. What are their particular recommendations? For example, education of elementary school students, high school students, or people who are preparing to take the driving test. Also, considering that many respondents do not have a first aid kit in the car, it should be stated whether it is mandatory by law or not.
Most references are up to-date and relevant to the topic, but it is necessary to standardize the journal titles (obey the instructions for authors), not to state the full journal name, and abbreviate it somewhere.
The manuscript generally contributes to improving public health, but it is necessary to emphasize why the results conducted on the Jordanian population are significant for the wider scientific community.
Comments on the Quality of English LanguageAuthor Response
Reviewer 1
Review of the manuscript: Knowledge, attitudes, and barriers to performing first aid during road accidents among the adult Jordanian population.
The manuscript covers a very important topic for public health. Namely, the study investigated the knowledge, attitudes, and barriers to performing first aid among the Jordanian population during road accidents.
Generally, the manuscript is technically correct, and the title fully corresponds to the research objective and topic; however, the manuscript requires some corrections.
-Thank you for your time and comments, which have improved the quality of our manuscript.
The introduction provides definitions of first aid and road traffic accidents. The authors also provide statistics on the frequency of traffic accidents in Jordan, but these are outdated. It would be good if they offered more recent data.
-The following has been added as suggested: ” Moreover, approximately 562 deaths and 17096 injuries were caused by traffic accidents in Jordan in 2022 alone.[7].”
In addition, the authors state why knowledge of first aid is important using the example of CPR, but these data date back to 1982. The authors can find confirmations for this claim in the available databases with significantly more recent references.
-This reference and the text associated with it have been removed and the following has been added:” A meta-analysis conducted in 2018 reported that bystander cardiopulmonary resuscitation enhances out-of-hospital cardiac arrest survival[10]. In a study conducted in the United States, bystander CPR was found to be associated with higher in-hospital survival rates among older patients with out-of-hospital cardiac arrest [11]. These findings highlight the life-saving potential of widespread BLS first-aid education[12].”
Given that the study was conducted in Jordan, I believe the authors should introduce potential readers to the fact that there is any if any, first aid training. For example, is it conducted at school or mandatory when taking a driving test or something similar? At the end of the introduction, the aim of the study is clearly defined. Namely, in this study, the authors wanted to investigate the Jordanian population's knowledge, attitudes, and barriers to performing first aid during road accidents.
-The following has been added to the discussion section: “Unfortunately, first-aid mandatory courses are not provided at schools or universities in Jordan. Moreover, although the driving theory test content in Jordan includes questions related to first aid, no practical training is offered.”
The method does contain all the necessary elements. The authors stated study design and setting, sample and data collection, instrument, and data analysis.
-Thank you for your comment.
The results are not clearly presented. It is necessary to correct the table for clarity. Insert a row labelled n and %, and then delete % from the cells in the table.
-Thank you for your comment; changes have been made as suggested.
However, only descriptive data is provided. Considering that the data analysis stated that a regression analysis was performed to determine predictors of knowledge and attitudes, it is necessary to add a table with those results.
-Thank you for your comment, regression tables have been added to the manuscript as suggested.
The conclusions are based on the obtained results. The conclusion is generally drawn based on the results. The authors should be more specific in their findings and discussion when recommending first aid training. What are their particular recommendations? For example, education of elementary school students, high school students, or people who are preparing to take the driving test. Also, considering that many respondents do not have a first aid kit in the car, it should be stated whether it is mandatory by law or not.
-The following has been added to the conclusion: “to include practical training as a requirement for obtaining a driving license and as mandatory courses for high school and university students.” And : ”Finally, as the results indicated that most drivers do not have a first-aid kit in their motor vehicles, implementing a law to make this mandatory could significantly impact life-saving efforts during road accidents. "
Most references are up to-date and relevant to the topic, but it is necessary to standardize the journal titles (obey the instructions for authors), not to state the full journal name, and abbreviate it somewhere.
-The corrections have been made as suggested.
The manuscript generally contributes to improving public health, but it is necessary to emphasize why the results conducted on the Jordanian population are significant for the wider scientific community
-The following was added to the Introduction: ’Due to its distinct social structure and geographic location, Jordan is an excellent case study for examining first aid knowledge and attitudes among the general population of the Middle East. The research's conclusions may improve our knowledge of public health readiness in these domains, increasing the study's importance and deepening our understanding of first aid knowledge and attitudes worldwide. Furthermore, improving first aid knowledge and attitudes is a common goal in public health. This study contributes to the global storage tank of information on public perception of first aid, ultimately helping in future public health education programs aiming at enhancing public knowledge and attitude with this regard.“ And: ’’Additionally, the study developed an Arabic tool to evaluate knowledge, attitudes, and barriers to performing first aid, which could be used in different Arabic-speaking countries.“.
Furthermore, the following has been added to the Conclusion: “The present findings are not only pertinent to Jordan but also relevant globally, as they reflect common challenges in first aid readiness that can significantly affect public health outcomes in emergencies. By establishing a detailed baseline of first-aid knowledge, attitudes, and barriers in Jordan, this study contributes to a broader understanding and highlights the critical need for targeted first-aid training interventions more broadly. Such insights are critical for designing evidence-based health policies and educational programs that can be adapted and implemented in various global contexts, ultimately enhancing the capacity to save lives in diverse populations.”
Reviewer 2 Report
Comments and Suggestions for Authors
Online based survey assessing knowledge, attitudes and barriers to perform first aid after road accidents conducted among adult Jordanians
Title consider adding 'an online survey'
Throughout the text you mention first-aid provider, CPR and basic life support - see for example different sections in the ERC guidelines with regard to these terms. In many guidelines these terms are distinct. Please describe what first aid includes according to your setting
Lines 57-58: This is a 45 years old study, anything more recent? If not mention this and say why
Who are your study subjects? How have they been selected? Dropout rates in your study? Show an algorithm of subjects included and lost along the study
Table 3. First aid for a broken limb
Lines 204-205 Content does not seem correct, ie that people with course had a lower knowledge than people without a course. Also, this is in contrast to Lines 257-258. Please amend, thank you
219-220: Below average knowledge compared to what? Add information please
Comments on the Quality of English LanguageFew minor mistakes, please check carefully
Author Response
Reviewer 2
-Thank you for time and comments, which have improved the quality of our manuscript.
Online-based survey assessing knowledge, attitudes, and barriers to perform first aid after road accidents conducted among adult Jordanians
Title consider adding 'an online survey'
-This has been modified as suggested.
Throughout the text you mention first-aid provider, CPR and basic life support - see for example different sections in the ERC guidelines with regard to these terms. In many guidelines these terms are distinct. Please describe what first aid includes according to your setting
- The following was added to the introduction“According to European Resuscitation Council Guidelines, first aid is defined as “the initial care provided for an acute illness or injury to preserve life, alleviate suffering, prevent further illness or injury and promote recovery” [5], while Basic life support BLS is more concerned with “initiating CPR in any unresponsive person with absent or abnormal breathing”. This study mainly focused on investigating first aid compo-nents[6].”
Lines 57-58: This is a 45 years old study, anything more recent? If not mention this and say why
- This reference and the text relevant to it was removed and the following was added: “A meta-analysis conducted in 2018 reported that bystander cardiopulmonary resuscitation enhances out-of-hospital cardiac arrest survival[10]. In a study conducted in the United States, bystander CPR was found to be associated with higher in-hospital survival rates among older patients with out-of-hospital cardiac arrest[11]. These findings highlight the life-saving potential of widespread BLS first-aid education[12]..”
Who are your study subjects? How have they been selected?
-The following was added to the manuscript:” An online questionnaire was developed on Google Forms based on a thorough literature review [19–21]. The questionnaire link and summary of the study objectives were distributed using various Jordanian social media platforms. Responses were collected between February and March 2024. Inclusion criteria consisted of being a Jordanian resident aged 18 or older. To confirm respondents met the inclusion criteria questions about place of residency, nationality and age were included in the questionnaire.” And:” Convenience and snowball sampling methods were adopted.”
Dropout rates in your study? Show an algorithm of subjects included and lost along the study
-As this is a cross-sectional study and not a longitudinal one, there were no dropouts, and because this is an online survey that was circulated in public social platform, response rates cannot be determined.
Table 3. First aid for a broken limb
-This has been corrected as suggested.
Lines 204-205 Content does not seem correct, ie that people with course had a lower knowledge than people without a course. Also, this is in contrast to Lines 257-258. Please amend, thank you
-This has been corrected as suggested.
219-220: Below average knowledge compared to what? Add information please
-Thank you for your comment. As stated in the manuscript, the median of the participants knowledge score was only 60%
Reviewer 3 Report
Comments and Suggestions for Authors
Dear Authors
Thanks for this questionnaire based study which is aim to investigate knowledge, attitudes, and barriers to performing first aid during 72 road accidents within the Jordanian population. Authors were prepare an online questionnaire and collected the data by using social media. The 732 volunteer admitted to the study and the median knowledge score regarding first aid items was 9, first aid attitude score was 24. The most important barrier to performing first aid was lack of the first aid training. In conclusion, the authors reported that there is a need for trainings on first aid. Because of the method there are some limitation at this study but the authors determined and described them at limitation section.
Thanks for this well written study.
Author Response
Dear Authors
Thanks for this questionnaire based study which is aim to investigate knowledge, attitudes, and barriers to performing first aid during 72 road accidents within the Jordanian population. Authors were prepare an online questionnaire and collected the data by using social media. The 732 volunteer admitted to the study and the median knowledge score regarding first aid items was 9, first aid attitude score was 24. The most important barrier to performing first aid was lack of the first aid training. In conclusion, the authors reported that there is a need for trainings on first aid. Because of the method there are some limitation at this study but the authors determined and described them at limitation section.
Thanks for this well written study.
-Thank you for your positive comment and encouraging words
Round 2
Reviewer 1 Report
Comments and Suggestions for Authors
The authors accept all the suggestions given, so the revised version of the manuscript titled Knowledge, attitudes, and Barriers to Performing First Aid During Road Accidents among the Adult Jordanian Populations is now easy to read, and all parts of the manuscript are better presented, which has improved the overall quality of the manuscript.
Considering those mentioned earlier and the importance of the topic the authors examined, I recommend that the manuscript be accepted after minor corrections. Namely, according to the reviewers’ recommendation, the authors added a table but did not mark it. So, the table with regression analysis does not have a number, title, or legend.
Author Response
The authors accept all the suggestions given, so the revised version of the manuscript titled Knowledge, attitudes, and Barriers to Performing First Aid During Road Accidents among the Adult Jordanian Populations is now easy to read, and all parts of the manuscript are better presented, which has improved the overall quality of the manuscript.
-Thank you for your comments and efforts that significantly improved the quality of the manuscript
Considering those mentioned earlier and the importance of the topic the authors examined, I recommend that the manuscript be accepted after minor corrections. Namely, according to the reviewers’ recommendation, the authors added a table but did not mark it. So, the table with regression analysis does not have a number, title, or legend
-All the table information were added